# *Waddlia chondrophila* and Male Infertility

**DOI:** 10.3390/microorganisms8010136

**Published:** 2020-01-17

**Authors:** David Baud, Nicolas Vulliemoz, Maria Verónica Morales Zapata, Gilbert Greub, Manon Vouga, Milos Stojanov

**Affiliations:** 1Materno-Fetal and Obstetrics Research Unit, Department Woman-Mother-Child, Lausanne University Hospital, 1011 Lausanne, Switzerland; David.Baud@chuv.ch (D.B.); Maria-Veronica.Morales-Zapata@chuv.ch (M.V.M.Z.); Manon.Vouga@chuv.ch (M.V.); 2Fertility Medicine and Gynaecologic Endocrinology Unit, Department Woman Mother Child, Lausanne University Hospital, 1011 Lausanne, Switzerland; Nicolas.Vulliemoz@chuv.ch; 3Institute of Microbiology, Lausanne University Hospital and University of Lausanne, 1011 Lausanne, Switzerland; Gilbert.Greub@chuv.ch

**Keywords:** *Waddlia*, chlamydia-like, male infertility, semen

## Abstract

*Waddlia chondrophila*, a *Chlamydia*-like bacterium, has been previously associated with adverse pregnancy outcomes. Analogously to *Chlamydia trachomatis*, *W. chondrophila* also negatively impacts human semen and may be a source of impaired male fertility. In this study, we analyzed *W. chondrophila* seroprevalence in a population of male patients of infertile couples and the impact of past exposition to this bacterium on semen parameters. Our results show a surprisingly high seroprevalence of *W. chondrophila*, which contrasts with a previous study focusing on a population of healthy men. Nevertheless, we did not observe any significant association between positive serology and abnormal sperm parameters. This may suggest that a negative impact on semen is observed only during an ongoing infection. Alternatively, *W. chondrophila* may have an immune impact on male fertility, as previously postulated for women with adverse pregnancy outcomes.

## 1. Introduction

Since the 1980s, a concerning decrease in semen quality has been reported, and potential causes are still being debated [1,2]. Etiologies, including endocrine and genetic disorders, have been demonstrated [3], but male infertility may be linked to environmental factors. Lifestyle habits, such as smoking, high BMI, and the increasing exposure to xenoestrogens, for example in drinking water, have been associated with reduced male fertility [4,5]. An additional hypothesis is that silent and/or chronic bacterial infections of the genital tract causing inflammation and direct deterioration of semen might partially explain some of the otherwise idiopathic cases [3,6,7,8]. The well-known intracellular pathogen, *Chlamydia trachomatis*, which can cause urethritis and negatively impact spermatozoa physiology, may be a key example [9,10,11,12].

Using an in vitro model of infection, we recently showed that *Waddlia chondrophila*, an emerging *Chlamydia*-like bacterium, has a negative impact on human spermatozoa [13]. By analogy with *C. trachomatis*, *W. chondrophila* was able to attach and penetrate into spermatozoa, reducing their viability and mitochondrial membrane potential, which is linked to motility. Moreover, standard sperm washing techniques were unable to completely eradicate it. 

*W. chondrophila* was first isolated from samples of aborted bovine fetuses [14,15]. This raised concerns for a potential abortigenic agent for other mammals, including humans. Three independent prospective serology based studies indicated that *W. chondrophila* might act as a possible inducer of miscarriage in pregnant women [16,17,18]. Moreover, *W. chondrophila* was shown to replicate in several human cell lines including fibroblasts, peripheral blood mononuclear cells, A549 pneumocytes, and in Ishikawa endometrial cells [14,19,20]. The seroprevalence of anti-*W. chondrophila* antibodies was significantly higher in women who experienced miscarriages compared to control groups of women with uneventful pregnancies. This association remained significant even after correction for age, ethnicity, and *C. trachomatis* serology status and was not due to cross-reactivity with other microorganisms known to induce pregnancy loss, such as *C. trachomatis*, *Coxiella burnetti*, *Toxoplasma gondii*, *Brucella abortus*, and *Parachlamydia acanthamoebae* [16]. Presence of *W. chondrophila* in human samples (placenta, vaginal swab, urine) was subsequently documented by PCR and/or immunohistochemistry [17]. 

Despite multiple reports of association with adverse pregnancy outcomes and the negative effect on human spermatozoa, it is not known whether *W. chondrophila* is associated with a deleterious effect on the male genital tract. This study aims to analyze whether this bacterium could be linked with reproductive disorders in men, namely men of infertile couples.

## 2. Material and Methods

### 2.1. Samples

Serum and sperm samples were collected from men of infertile couples seeking infertility investigations at the Fertility Medicine Unit (Lausanne University Hospital, Lausanne, Switzerland). This study was carried out in accordance with the recommendations of the Cantonal Human Research Ethics Commission of Vaud (CER-VD)( protocol 265/14), according to the Swiss Federal Act on Research involving Human Beings. The study protocol was approved by the commission (protocol 265-14). All patients were fully informed about the content of the research project and gave their written consent to be included in the study.

Semen assessment was performed in the Laboratory of Andrology and Reproductive Biology (LABR, Lausanne University Hospital, Switzerland). Semen was obtained by masturbation after 2 to 5 days of sexual abstinence. After a 30 min liquefaction at 37 °C, samples were manually assessed for volume, pH, and morphology using the Papanicolaou method. CASA SCA (Version 5.4, Microptic SL, Barcelona, Spain) computer-assisted sperm analysis software was used to evaluate sperm concentration and motility (total and progressive). All analyses were performed following the 2010 World Health Organization laboratory manual for the examination and processing of human semen guidelines (5th edition).

### 2.2. Microimmunofluorescence

Serology for *W. chondrophila* was performed as described previously [16,21]. Briefly, formalin-inactivated *W. chondrophila* strain ATCC VR-1470 was deposited on serology slides as an antigen. Serum samples were serially diluted in phosphate-buffered saline, and dilutions of 1/32 and 1/64 were included in the analysis. Mouse anti-human IgG secondary antibody conjugated to Alexa Fluor 488 dye (Thermo Fisher Scientific, Allschwil, Switzerland, diluted 1/1000) was used to detect the presence of anti-*W. chondrophila* IgG.

Microimmunofluorescence tests were analyzed by two independent evaluators. Weighted Cohen’s kappa coefficient was determined for each series to assess the agreement of the results and always ranked between “good” and “very good”. In cases of discrepancy, a third evaluator was involved, and the case was discussed to reach consensus.

### 2.3. Enzyme-Linked Immunosorbent Assay (ELISA)

Serology for *C. trachomatis* was performed with the “Anti-*Chlamydia trachomatis* ELISA (IgG)” (EI 2191-9601 G, EUROIMMUN Schweiz AG, Luzern, Switzerland), according to the manufacturer’s indications.

### 2.4. DNA Extraction from Sperm Samples and Real-Time Quantitative Polymerase Chain Reaction Assay (qPCR)

DNA was extracted from 1 mL of semen using the QIAamp DNA mini kit (Qiagen AG, Basel, Switzerland) following the manufacturer’s specifications with the addition of 43 mM DTT to the lysis buffer as the sole modification. All samples were analyzed using a *W. chondrophila*-specific qPCR and a Chlamydiales-specific qPCR, as previously described [22,23].

### 2.5. DNA Sequencing and Analysis

DNA sequencing was performed by Eurofins Genomics (Eurofins Genomics GmbH, Kostanz, Germany), with the primers used for the Chlamydiales-specific qPCR [23]. Obtained sequences were manually trimmed and blasted against the NCBI nr/nt database (partially non-redundant nucleotide sequences from GenBank, EMBL, and DDB, analysis performed on September 2019) using the blast module of the Geneious prime software (version 2020.1, Biomatters Limited, Auckland, New Zealand). For each positive sample, the top hit from the blast analysis was presented in Table 1.

### 2.6. Statistical Analysis

Statistical analysis was performed with RStudio Version 1.2.1335 (RStudio, Inc., Boston, MA, USA). Categorical variables were compared using the Pearson χ2 test, while continuous variables were compared using the Wilcoxon–Mann–Whitney test. A *p*-value < 0.05 was considered statistically significant.

## 3. Results

A total of 204 men of infertile couples were enrolled in the study, and their semen parameters were evaluated using the WHO guidelines. The impact of *W. chondrophila* on semen parameters was assessed by performing serological analysis on the serum samples. A total of 58.3% of patients had *W. chondrophila*-specific IgG, indicating a previous exposure to the microorganism. Our results indicated that positive serology was not significantly associated with abnormal semen status (Table 2 and Table 3). In addition, seropositivity was not linked to a reduced value of specific spermiogram parameters, including total spermatozoa count and concentration, total motility, progressive motility, and morphology (Table 2). Similar results were observed for *C. trachomatis*, although seroprevalence was lower in this case (Table 2). Interestingly, we observed an association with patient age, as the group with positive serology (1/32) was significantly older compared to men with negative serology (Table 2).

Given the high exposure to the bacterium, we wanted to evaluate whether *W. chondrophila* actively colonizes sperm of patients included in the study. Total DNA was extracted from semen samples and analyzed with a specific qPCR, which indirectly shows presence of the bacteria in the sample. None of the samples tested positive for *W. chondrophila* DNA. For this reason, we did not attempt the isolation of the bacterium from semen. 

To analyze the presence of *Chlamydia*-like bacteria more broadly, samples were additionally tested with a Chlamydiales-specific qPCR. A total of 2.9% (*n* = 6) of samples were positive with qPCR analysis, and their amplified DNA was purified, sequenced, and compared to sequences present in nucleotide sequence databases. Top hits obtained by blast analysis with the highest identity and alignment lengths are shown in Table 1. Interestingly, five of these six patients had abnormal semen parameters.

## 4. Discussion

Multiple serological studies have demonstrated an association of the intracellular bacterium *W. chondrophila* with adverse pregnancy outcomes. Most studies have targeted female patients, while seroprevalence of *W. chondrophila* in men has only been investigated on two occasions, until now. In the first study, 13.7% of healthy young Swiss men (66 out of 482) attending military service tested positive [24], while in the second, this was the case for 45% (18 out of 40) of Israeli men [25]. 

We have previously shown that human spermatozoa exposed to *W. chondrophila* displayed a reduction of viability and mitochondrial membrane potential [13]. Therefore, this study focused on a population of men of infertile couples in order to address whether exposure to this bacterium negatively impacts male fertility.

Interestingly, we observed a high seroprevalence of 58.3%, which is higher compared to all previous serology studies, including those with women who experienced adverse pregnancy outcomes [16,17,18]. On the contrary, we did not detect *W. chondrophila* DNA, indicating that this bacterium might not colonize semen. The high seroprevalence and absence by qPCR detection may be explained by multiple scenarios. For example, (a) infection may have occurred in the past, and thus the qPCR is negative; (b) the infection is ongoing, but in a body site other than the male reproductive system; (c) the infection occurred in the past in another body site.

One important limitation of our study is the absence of a control population consisting of men with proven fertility, which was not explored due to ethical aspects and the difficulty of obtaining semen in healthy individuals. Moreover, we cannot rule out the presence of PCR inhibitors in DNA extracted from semen, which could inhibit the *W. chondrophila*-specific qPCR. Nevertheless, our previous study showed that this bacterium could be detected within artificially infected sperm samples by the same qPCR [13].

Similar studies performed with *C. trachomatis*, which was also included in our study, gave highly variable outcomes [26,27,28]. It is still not clear if and how *C. trachomatis* impacts male fertility, although it can cause epididymitis, urethritis, and prostatitis, all of which might lead to an impairment of reproductive potential [3,29]. Moreover, it can have a direct negative effect on spermatozoa physiology [10,30,31,32], as shown in vitro for *W. chondrophila* [13]. In the present clinical study, we did not observe any significant association with abnormal semen parameters in males that tested positive for *W. chondrophila*. Male seropositivity was not associated with adverse pregnancy outcomes such as miscarriage in their female partners. 

Although we did not find *W. chondrophila* in the semen of patients included in this study, we detected the presence of Chlamydiales DNA in 2.4% of samples (*n* = 6). Of note, the majority of these patients (*n* = 5), had abnormal spermiograms. Given that the detected sequences belonged to previously uncultured Chlamydiales, the role of these isolates in semen colonization warrants further investigation, including isolation and assessment of the impact on spermatozoa physiology. 

Interestingly, *W. chondrophila* serology was significantly higher in patients above the age of 30, meaning that the risk of infection may increase with age. Compared to our previous study on healthy Swiss males, in which the mean age was 20.6 ± 1.4 and *W. chondrophila* seropositivity was 13.7% [24], the high seroprevalence observed in men of infertile couples may suggest a deleterious immunological role of *W. chondrophila* on male fertility. Similarly to what has been described earlier [21], the immunological response against *W. chondrophila* may involve the induction of an autoimmune response against an epitope shared by the bacteria and spermatozoa/ova antigens. Alternatively, a transient negative effect on male infertility may be observed only during an ongoing infection.

Overall, the high *W. chondrophila* seroprevalence indicates frequent exposure of men to this bacterium in Western Switzerland. Therefore, further studies are required to understand the impact of this bacterium on human health.

## Figures and Tables

**Table 1 microorganisms-08-00136-t001:** Chlamydiales bacteria detected in the semen samples.

Sample	Hit	Accession Number	Identity (%)	Alignment Length (bp)	*W. chondrophila* Serology	*C. trachomatis* Serology	Spermiogram
1	Uncultured Chlamydiales bacterium clone 14-41	KX451108	96.8	156	Pos	Neg	Abnormal
7	Uncultured Chlamydiales bacterium clone 12-15	KX451048	95.7	162	Pos	Neg	Abnormal
10	Uncultured Chlamydiales bacterium clone 14-02	JX083073	91.4	175	Pos	Neg	Abnormal
135	Uncultured Chlamydiales bacterium clone HE210001biof	JX083106	94.9	78	Pos	Neg	Normal
222	Uncultured Chlamydiales bacterium clone AHDr12	JQ860021	100	181	Pos	Neg	Abnormal
239	Uncultured Chlamydiales bacterium clone AHDr12	JQ860021	100	181	Pos	Neg	Abnormal

**Table 2 microorganisms-08-00136-t002:** Characteristics of patients according to *W. chondrophila* serology.

Characteristics	*W. chondrophila* Serology 1/32			*W. chondrophila* Serology 1/64		
	Negative *n* = 85	Positive *n* = 119	*p*-Value	Negative *n* = 106	Positive *n* = 98	*p*-Value
**Demographic characteristics**						
Age (years)	35.9 ± 7.0	38.0 ± 6.1	0.005	36.9 ± 7.2	37.4 ± 5.8	0.322
<30	20	11	0.005	20	11	0.185
>30	65	108		86	87	
Origin						
Swiss	39	58	0.687	52	45	0.758
Non-swiss	46	61		54	53	
Origin partner						
Swiss	33	38	0.308	41	30	0.289
Non-swiss	52	81		65	68	
Education						
University	21	29	0.956	28	22	0.621
Non-university	64	90		78	76	
Place of residence						
City	66	81	0.133	77	70	0.971
Rural	19	38		29	28	
Pets						
Yes	26	38	0.838	36	28	0.498
No	59	81		70	70	
**Serology (IgG)**						
*C. trachomatis* serology						
Neg	72	98	0.657	88	82	0.950
Pos	13	21		18	16	
**Semen characteristics**						
Spermiogram						
Normal	18	29	0.593	24	23	0.888
Abnormal	67	90		82	75	
Sperm concentration (× 10^6^/mL)						
Mean	37.53	39.88	0.862	38.59	39.24	0.815
SD	42.85	39.48		41.69	40.09	
Total sperm count (× 10^6^)						
Mean	112.06	116.31	0.867	109.7	119.77	0.879
SD	115.75	122.11		110.26	128.6	
Total motility (%)						
Mean	53.44	52.69	0.841	52.82	53.19	0.864
SD	23.22	23.74		23.45	23.6	
Progressive motility (%)						
Mean	37.65	36.44	0.775	37.26	36.59	0.992
SD	18.77	18.12		18.86	17.88	
Sperm morphology (%)						
Mean	2.52	2.91	0.343	2.61	2.89	0.546
SD	2.27	2.51		2.31	2.53	
**Clinical history**						
Andrological issues						
Yes	9	13	0.939	11	11	0.845
No	76	106		95	87	
Obstetrical issues (partner)						
Yes	35	37	0.137	44	28	0.074
No	50	82		62	70	
Miscarriages (partner)						
Yes	32	32	0.103	39	25	0.241
No	53	87		67	73	

**Table 3 microorganisms-08-00136-t003:** Characteristics of patients according to semen parameters.

Characteristics	Sperm Parameters
Normal*n* = 47	Abnormal*n* = 157	*p*-Value
**Demographic characteristics**			
Age (years)	37.2 ± 7.2	37.1 ± 6.4	0.899
<30	8	23	0.691
>30	39	134	
Origin			
Swiss	30	67	0.011
Non-swiss	17	90	
Origin partner			
Swiss	20	51	0.204
Non-swiss	27	106	
Education			
University	15	35	0.179
Non-university	32	122	
Place of residence			
City	36	111	0.429
Rural	11	46	
Pets			
Yes	17	47	0.419
No	30	110	
**Serology**			
*W. chondrophila* serology			
Neg	18	67	0.593
Pos	29	90	
*C. trachomatis* serology			
Neg	37	133	0.334
Pos	10	24	
**Semen characteristics**			
Sperm concentration (× 10^6^/mL)			
Mean	80.43	26.47	<0.001
SD	43.43	30.63	
Total perm count (× 10^6^)			
Mean	231.69	79.47	<0.001
SD	123.05	93.06	
Total motility (%)			
Mean	75.77	46.18	<0.001
SD	12.02	21.71	
Progressive motility (%)			
Mean	54.85	31.58	<0.001
SD	11.55	16.54	
Sperm morphology (%)			
Mean	5.47	1.93	<0.001
SD	1.8	1.93	
**Clinical history**			
Andrological issues			
Yes	5	17	0.971
No	42	140	
Obstetrical issues (partner)			
Yes	20	52	0.235
No	27	105	
Miscarriages (partner)			
Yes	31	109	0.653
No	16	48

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
