# Peer review of "Waddlia chondrophila and Male Infertility"

_microorganisms, 2020, doi:10.3390/microorganisms8010136_

Round 1

Reviewer 1 Report

In the submitted manuscript D. Baud et al describes the potential impact of Waddlia chondrophila infection on male fertility. They tested serum and semen samples for the presence of W. chondrophila IgG seropositivity and W. chondrophila DNA. They concluded that the IgG seropositivity was not correlated to the abnormal spermiogram, and they also could not find W. chondrophila DNA in the semen samples, but in 6 samples they found the DNA of other Chlamydiales bacteria.  The lack of W. chondrophila DNA does not support the role of this pathogen in abnormal spermium functions, and supported the fact that there was no acute/ongoing infection in any of the patients. The authors hypothesized that autoimmune mechanisms against shared bacterial/human epitopes could be at the background of the altered spermium functions.

Altogether the manuscript contains grossly negative data, which is up to the Journal’s policy to be accepted or not. There are some improvements could also be made.

Instead of +/- seropositivity can the authors provide some quantitative/semiquantitative evaluation (e.g seropositive both in the 32 and 64 fold dilutions, or strongly-medium-weakly seropositive in a certain dilution). May be the strongly seropositive patients show some spermium malfunctions. The designation of Chlamydiales specific RT-PCR should be corrected to qPCR in row 108. The RT-PCR designation suggests that it detected RNA which was not the case. An interesting information is the detection of Chlamydiales DNA in 6 samples. Can the authors describe these samples in detail (as in Table 1)? Are these patients have some special anamnestic or spermium characteristics?

Author Response

In the submitted manuscript D. Baud et al describes the potential impact of Waddlia chondrophila infection on male fertility. They tested serum and semen samples for the presence of W. chondrophila IgG seropositivity and W. chondrophila DNA. They concluded that the IgG seropositivity was not correlated to the abnormal spermiogram, and they also could not find W. chondrophila DNA in the semen samples, but in 6 samples they found the DNA of other Chlamydiales bacteria.  The lack of W. chondrophila DNA does not support the role of this pathogen in abnormal spermium functions, and supported the fact that there was no acute/ongoing infection in any of the patients. The authors hypothesized that autoimmune mechanisms against shared bacterial/human epitopes could be at the background of the altered spermium functions.

Altogether the manuscript contains grossly negative data, which is up to the Journal’s policy to be accepted or not. There are some improvements could also be made.

We agree with the reviewer’s summary of our study. Despite negative data presented here (no association of W. chondrophila DNA or serology with abnormal spermiogram), we identified:

A high chondrophila seroprevalence in men from infertile couple 2.3% of the sperm samples contain other Chlamydiales bacteria

We think that these results still warrants publication and further investigation, as well as the negative association presented here.

Instead of +/- seropositivity can the authors provide some quantitative/semiquantitative evaluation (e.g seropositive both in the 32 and 64 fold dilutions, or strongly-medium-weakly seropositive in a certain dilution). May be the strongly seropositive patients show some spermium malfunctions.

We have included the W. chondrophila serology results of the 1/64 dilution, which was performed in parallel with the 1/32 dilution (revised Table 1). As observed, even if taking in account a higher threshold of seropositivity, W. chondrophila positive serology was still not associated with abnormal spermiogram values.

The designation of Chlamydiales specific RT-PCR should be corrected to qPCR in row 108. The RT-PCR designation suggests that it detected RNA which was not the case. An interesting information is the detection of Chlamydiales DNA in 6 samples. Can the authors describe these samples in detail (as in Table 1)? Are these patients have some special anamnestic or spermium characteristics?

We have corrected the term RT-PCR. Moreover, we have updated Table 3 (formerly Table 2) with additional information. Interestingly, 5 out of 6 patients in which Chlamydiales bacteria were detected, have an abnormal spermiogram. This was also added in the result section.

Reviewer 2 Report

This study investigated the bacterial species Waddlia chondrophila. This species has been associated with adverse pregnancy outcomes in mammals, but has not been extensively investigated for its impact upon male fertility. The authors had previously conducted in vitro investigations on interactions between the bacteria and sperm. In this study, they expand on this work to look at samples from male patients with fertility concerns.

Major concerns

It is curious that the Introduction does not mention the levels of oral contraceptive that are in drinking water, which has clearly been linked to issues with male fertility in many areas. Although this is not linked to anything microbiological, it is a known issue that should be mentioned (and sources cited) and the potential for synergy with the current study could have been considered.

The results are short and a bit hard to follow in how they are explained but it would seem that more than half of the patient samples were IgG seropositive for W. chondrophila, which would suggest at least a past exposure to the bacteria, if not a current infection. However, the specificity of the microimmunofluorescence is not demonstrated (and it is not clear if this was done in the publications referred to as ‘previously’, references 14 & 19). If the serum is able to non-specifically bind to an antigen that is in common with other bacterial species, it may not be indicative of current or past W. chondrophila infection in the patients.

Once this is resolved, it should be discussed in the context of the difference between the immunoassay and the qPCR results. A result suggesting over 50% and another saying no W. chondrophila certainly suggests one or both assays need refinement, discussion, and carefully included controls.

It is curious why there was no attempt to culture W. chondrophila from the semen samples and to then attempt to correlate abnormal semen parameters with positive cultures. This would have been a clear follow-on from the previous in vitro work, however perhaps there was some reason that this was not feasible (perhaps transport time or issues with culturing from the patient samples). It would be interesting to hear in the manuscript if this was attempted and the outcome(s).

Minor concerns

Line 18: The ‘a’ and ‘with’ are swapped. It should read ‘with a’ and not ‘a with’.

Lines 126-7: It is not clear why collection of semen from healthy individuals was an ethical concern.

Table 1 is hard to follow. The various subsections with the parameters such as age, education, pets, etc. need to have some short of division line to make it clearer what is happening with the data. Some footnotes to better clarify the meaning of the column headers would be appropriate as well.

Table 2 presumably are sequences of the qPCR results (this should be clearer in the text). It should be made clearer in the text and here the expected fragment size of the qPCR product.

Author Response

This study investigated the bacterial species Waddlia chondrophila. This species has been associated with adverse pregnancy outcomes in mammals, but has not been extensively investigated for its impact upon male fertility. The authors had previously conducted in vitro investigations on interactions between the bacteria and sperm. In this study, they expand on this work to look at samples from male patients with fertility concerns.

Major concerns

It is curious that the Introduction does not mention the levels of oral contraceptive that are in drinking water, which has clearly been linked to issues with male fertility in many areas. Although this is not linked to anything microbiological, it is a known issue that should be mentioned (and sources cited) and the potential for synergy with the current study could have been considered.

We acknowledge the importance of environmental factors other than “microbiological” ones. We added the following sentence in the introduction: “Lifestyle habits, such as smoking, high BMI and the increasing exposure to xenoestrogens, for example in drinking water, have been associated with reduced male fertility [4,5].”.

The results are short and a bit hard to follow in how they are explained but it would seem that more than half of the patient samples were IgG seropositive for W. chondrophila, which would suggest at least a past exposure to the bacteria, if not a current infection. However, the specificity of the microimmunofluorescence is not demonstrated (and it is not clear if this was done in the publications referred to as ‘previously’, references 14 & 19). If the serum is able to non-specifically bind to an antigen that is in common with other bacterial species, it may not be indicative of current or past W. chondrophila infection in the patients.

Once this is resolved, it should be discussed in the context of the difference between the immunoassay and the qPCR results. A result suggesting over 50% and another saying no W. chondrophila certainly suggests one or both assays need refinement, discussion, and carefully included controls.

The two techniques answer different questions:

1) Serology analysis shows the presence/absence of specific IgG antibodies against W. chondrophila and indicate whether an individual has come in contact with the microorganism previously. Positive and negative controls were always included in our serological analysis.

Specificity of this method has previously been demonstrated, without any cross-reactivity to closely related species, including Chlamydia trachomatis, Chlamydia pneumoniae, and Chlamydia psittaci; or other pathogen known to induce adverse pregnancy outcomes. This was described in the introduction of our paper: “… was not due to cross-reactivity with other microorganisms known to induce pregnancy loss, such as C. trachomatis, Coxiella burnetti, Toxoplasma gondii, Brucella abortus and Parachlamydia acanthamoebae [16].

2) qPCR demonstrate the presence of the pathogen in a given sample, sperm in our case. DNA amplification is a highly sensitive method (see next reviewer question). This does not exclude the presence and infection by W. chondrophila in another body site

The high seroprevalence and absence by qPCR can be explained by multiple scenarios. For example (a) Infection may occur in the past (positive IgG) and thus the qPCR remain negative (no ongoing infection); (b) The infection is ongoing, but in body site other than the male reproductive system; (c) The infection occurred in the past in another body site.

It is curious why there was no attempt to culture W. chondrophila from the semen samples and to then attempt to correlate abnormal semen parameters with positive cultures. This would have been a clear follow-on from the previous in vitro work, however perhaps there was some reason that this was not feasible (perhaps transport time or issues with culturing from the patient samples). It would be interesting to hear in the manuscript if this was attempted and the outcome(s).

Indeed, one of our goal was to isolate W. chondrophila from human sperm. However, absence of W. chondrophila DNA indicated the absence of this bacterium in the samples (qPCR being a highly sensitive method for the investigation of the presence of the bacteria, see [22]). Of note, detection by qPCR has a higher sensitivity to detect the bacterium compared to isolation by cell culturing. Moreover, we used 1 ml of sperm to perform DNA extraction, which is higher to the volume that would have been inoculated on cell lines to culture the bacteria. We have specified that isolation of W. chondrophila from semen was not performed.

Minor concerns

Line 18: The ‘a’ and ‘with’ are swapped. It should read ‘with a’ and not ‘a with’.

This was corrected.

Lines 126-7: It is not clear why collection of semen from healthy individuals was an ethical concern.

The collection of semen from healthy individuals was not accepted by our Ethical committee and therefore was not used in the present study. Moreover, healthy volunteer would have been difficult to recruit and to convince to participate to our study, since they do not spontaneously present to any clinic of our hospital.

Table 1 is hard to follow. The various subsections with the parameters such as age, education, pets, etc. need to have some short of division line to make it clearer what is happening with the data. Some footnotes to better clarify the meaning of the column headers would be appropriate as well.

We agree with reviewer that the table is complex. We created a new Table 2 containing patients’ characteristics according to semen parameters, while Table 1 contains now only patients’ characteristics according to W. chondrophila serology. Moreover, we added headings and divisions for each section of the two tables and added some specifications for the semen parameters. We hope that this will help the understanding of the tables.

Table 2 presumably are sequences of the qPCR results (this should be clearer in the text). It should be made clearer in the text and here the expected fragment size of the qPCR product.

Table 3 (formerly Table 2) contains the top hit obtained by BLAST, as well as the length of the alignment. We have modified the text and we hope that the information is now clearer.

Round 2

Reviewer 1 Report

Since 5 of the 6 Chlamydiales  qPCR positive samples had abnormal spermiogram, which seems to be pretty significant, I feel that it could have been discussed in the Discussion section. There is no discussion about it in the revised form of the manuscript.

Author Response

We agree with the reviewer that this interesting finding should be further discussed in the Discussion section. Therefore, we changed the corresponding paragraph as follows (Lines 192-195):

“Although we did not find W. chondrophila in the semen of patients included in this study, we detected the presence of Chlamydiales DNA in 2.4% of samples (n=6). Of note, the majority of these patients (n=5), had an abnormal spermiogram. Given that the detected sequences belonged to previously uncultured Chlamydiales, the role of these isolates in semen colonization warrants further investigation, including isolation and assessment of the impact on spermatozoa physiology.”

Reviewer 2 Report

In the original review, three major concerns were noted: one in the Introduction; a set of associated concerns in the Results; and one related to culturing. The Introduction one has been addressed and the response clearly indicates where in the revised manuscript a change has been made, which was highlighted in the revision. The other two are either not addressed or not sufficiently addressed.

The Results remain short and difficult to follow - a major concern that has not been addressed. It therefore remains difficult to the average reader to understand the two methods, which the authors have here explained in their response to the reviewers, but not to their future readers. Readers will need to know why the serology analyses were done and their expected outcomes and the ramifications for prior patient exposures. Readers will need to know the multiple scenarios for the qPCR, which are quite clearly listed here in this response - but will never be seen by the wider public! This needs to be in the manuscript and in general the writing needs to be clearer and explain things more completely to the reader with less assumed knowledge (such as the work that was done previously to show the lack of cross-reactivity). The phrasing in the rebuttal about cross-reactivity is also curiously specific, which suggests maybe there is some cross-reactivity, just not to species involved in fertility?

Regarding culturing - there are numerous instances in the literature where samples are PCR or qPCR negative, yet culture positive and vice versa. It is known that biological samples have many inhibitors of PCR that can interfere with generation of products and since you have no positive controls, not attempting to culture from the samples - as in the yellow highlight in the revision, is not sufficient justification. There should be a caveat somewhere that the qPCR may have been inhibited.

Some of the minor concerns need to be better addressed as well. Original lines 126-7 about ethics needs to be phrased better, since this should have been possible; most ethics committees would have allowed it and recruitment of healthy volunteers is possible even at hospital clinics. Whilst we appreciate the effort of making Table 1 into two to try to make it easier to read, it is still difficult to pull out the information under the Demographic Characteristics. You have centre justified everything in the first column, making it hard to see the different Origins from the name Origin. Perhaps the titles of categories themselves should be left justified and the categories could remain centre justified to resolve this. Clarification on the data in Table 3 is good, however the Methods only mention sequencing in 2.5, not how you did the BLAST search (in fact, BLAST is not mentioned in the paper, only here in the response to reviewers) or the database interrogated. This is not mentioned in the Results either.

Author Response

In the original review, three major concerns were noted: one in the Introduction; a set of associated concerns in the Results; and one related to culturing. The Introduction one has been addressed and the response clearly indicates where in the revised manuscript a change has been made, which was highlighted in the revision. The other two are either not addressed or not sufficiently addressed.

The Results remain short and difficult to follow - a major concern that has not been addressed. It therefore remains difficult to the average reader to understand the two methods, which the authors have here explained in their response to the reviewers, but not to their future readers. Readers will need to know why the serology analyses were done and their expected outcomes and the ramifications for prior patient exposures.

 As requested by the reviewer, we have added our previous responses in the text. We hope that the new version of the results section is clearer and that it will guide even the non-experimented reader through the results that we obtained. Furthermore, results are widely discussed in the corresponding section.

Readers will need to know the multiple scenarios for the qPCR, which are quite clearly listed here in this response - but will never be seen by the wider public! This needs to be in the manuscript and in general the writing needs to be clearer and explain things more completely to the reader with less assumed knowledge (such as the work that was done previously to show the lack of cross-reactivity). The phrasing in the rebuttal about cross-reactivity is also curiously specific, which suggests maybe there is some cross-reactivity, just not to species involved in fertility?

We added a more critical discussion of our study, listing the multiple scenarios of high seroprevalence and absence of qPCR detection. Moreover, we listed the main limitations, including the absence of a control population (Lines 144-148).

In our previous studies, we tested possible cross-reactivities to all W. chondrophila-closely related bacteria (other Chlamydiales) and pathogens involved in fertility/obstetric (such as Toxoplasma). None of these pathogens showed any cross-reactivity with Waddlia.

 Regarding culturing - there are numerous instances in the literature where samples are PCR or qPCR negative, yet culture positive and vice versa. It is known that biological samples have many inhibitors of PCR that can interfere with generation of products and since you have no positive controls, not attempting to culture from the samples - as in the yellow highlight in the revision, is not sufficient justification. There should be a caveat somewhere that the qPCR may have been inhibited.

This issue was listed as a limitation of the study. In the section about the limitations, we added:

Moreover, we cannot rule out the presence of PCR inhibitors in DNA extracted from semen, which could inhibit the W. chondrophila-specific qPCR. Nevertheless, our previous study showed that this bacterium could be detected within artificially infected sperm sample by the same qPCR [13].” (Lines 151-154)

 Some of the minor concerns need to be better addressed as well. Original lines 126-7 about ethics needs to be phrased better, since this should have been possible; most ethics committees would have allowed it and recruitment of healthy volunteers is possible even at hospital clinics.

This sentence was rephrased and listed as a limitation of our study:

“One important limitation of our study is the absence of a control population consisting of men with proven fertility, which was not explored due to ethical aspects and the difficulty of obtaining semen in healthy individuals.” (Lines 149-151)

Whilst we appreciate the effort of making Table 1 into two to try to make it easier to read, it is still difficult to pull out the information under the Demographic Characteristics. You have centre justified everything in the first column, making it hard to see the different Origins from the name Origin. Perhaps the titles of categories themselves should be left justified and the categories could remain centre justified to resolve this.

We agree with the reviewer: unfortunately, the formatting performed after submission changed our modifications. We hope that the actual formatting of the tables facilitates their comprehension.

 Clarification on the data in Table 3 is good, however the Methods only mention sequencing in 2.5, not how you did the BLAST search (in fact, BLAST is not mentioned in the paper, only here in the response to reviewers) or the database interrogated. This is not mentioned in the Results either.

We added the following paragraph in the Material and methods section:

“Obtained sequences were manually trimmed and blasted against the NCBI nt database (partially non-redundant nucleotide sequences from GenBank, EMBL, and DDB) using the blast module of the Geneious prime software (Biomatters Limited, Auckland, New-Zealand). For each positive sample, the top hit from blast analysis was presented in table 3.” (Lines 93-97)

 We added in the result section the following clarification:

“Top hits obtained by blast analysis with the highest identity and alignment lengths are shown in Table 3.” (Lines 122-123)

Round 3

Reviewer 2 Report

The changes to the manuscript and tables have much improved its clarity, making it more accessible to the reader. The additional material and better formatting of data in the tables are significant improvements on the presentation of the information and the discussion of the study. There remain only a few minor details from the new information that has been added, which the authors can quickly address.

Minor changes

Methods:

For reproducibility, the authors may wish to include the approximate date (month and year) of the blast searches so that the readers and anyone wishing to reproduce the work will know why their blast results may differ. (This is optional, but often now included in manuscripts.) Additionally, the version of the Geneious software must be included.

Results:

In line 105, it is suggested that the authors change to “The impact of…” to correct the grammar.

In line 106, it is suggested that the authors change to “on the serum samples.” to correct the grammar.

Author Response

The changes to the manuscript and tables have much improved its clarity, making it more accessible to the reader. The additional material and better formatting of data in the tables are significant improvements on the presentation of the information and the discussion of the study. There remain only a few minor details from the new information that has been added, which the authors can quickly address.

We thank reviewer 2 for the significant improvement of our manuscript. The latest modifications are highlighted in yellow in the new version.

Minor changes

Methods:

For reproducibility, the authors may wish to include the approximate date (month and year) of the blast searches so that the readers and anyone wishing to reproduce the work will know why their blast results may differ. (This is optional, but often now included in manuscripts.) Additionally, the version of the Geneious software must be included.

We added the information concerning the version of the Geneious software and when blast analysis was performed (Lines 95-96).

Results:

In line 105, it is suggested that the authors change to “The impact of…” to correct the grammar.

In line 106, it is suggested that the authors change to “on the serum samples.” to correct the grammar.

The two sentences were corrected (Lines 105 and 106).